# H3K4me3, H3K9ac, H3K27ac, H3K27me3 and H3K9me3 Histone Tags Suggest Distinct Regulatory Evolution of Open and Condensed Chromatin Landmarks

**DOI:** 10.3390/cells8091034

**Published:** 2019-09-05

**Authors:** Anna A. Igolkina, Arsenii Zinkevich, Kristina O. Karandasheva, Aleksey A. Popov, Maria V. Selifanova, Daria Nikolaeva, Victor Tkachev, Dmitry Penzar, Daniil M. Nikitin, Anton Buzdin

**Affiliations:** 1Mathematical Biology & Bioinformatics Laboratory, Institute of Applied Mathematics and Mechanics, Peter the Great St.Petersburg Polytechnic University, Polytechnicheskaya 29, St. Petersburg 195251, Russia; 2Laboratory of Microbiological Monitoring and Bioremediation of Soil, All-Russia Research Institute for Agricultural Microbiology, Podbel’skogo, 3, St. Petersburg 196608, Russia; 3Lomonosov Moscow State University, Vorobiovy Gory 1, Moscow 119991, Russia; 4Research Centre for Medical Genetics, Moskvorechie Street 1, Moscow 115478, Russia; 5Omicsway Corp., Walnut, CA 91789, USA; 6Vavilov Institute of General Genetics Russian Academy of Sciences, Gubkina 3, Moscow 119991, Russia; 7Shemyakin-Ovchinnikov Institute of Bioorganic Chemistry, Moscow 117997, Russia; 8I.M. Sechenov First Moscow State Medical University, Moscow 119991, Russia

**Keywords:** human genome molecular evolution, histone modifications, transposable elements, retrotransposons, molecular pathway analysis, gene ontology, epigenetics, gene regulation, promoter, chromatin structure

## Abstract

Background: Transposons are selfish genetic elements that self-reproduce in host DNA. They were active during evolutionary history and now occupy almost half of mammalian genomes. Close insertions of transposons reshaped structure and regulation of many genes considerably. Co-evolution of transposons and host DNA frequently results in the formation of new regulatory regions. Previously we published a concept that the proportion of functional features held by transposons positively correlates with the rate of regulatory evolution of the respective genes. Methods: We ranked human genes and molecular pathways according to their regulatory evolution rates based on high throughput genome-wide data on five histone modifications (H3K4me3, H3K9ac, H3K27ac, H3K27me3, H3K9me3) linked with transposons for five human cell lines. Results: Based on the total of approximately 1.5 million histone tags, we ranked regulatory evolution rates for 25075 human genes and 3121 molecular pathways and identified groups of molecular processes that showed signs of either fast or slow regulatory evolution. However, histone tags showed different regulatory patterns and formed two distinct clusters: promoter/active chromatin tags (H3K4me3, H3K9ac, H3K27ac) vs. heterochromatin tags (H3K27me3, H3K9me3). Conclusion: In humans, transposon-linked histone marks evolved in a coordinated way depending on their functional roles.

## 1. Introduction

Transposons are endogenous mobile components of a genome that can replicate themselves into new genomic locations [1]. Retroelements (REs) form a specific group of transposons: they proliferate through cDNA intermediates generated with reverse transcription of their RNA transcripts [2]. REs are considered to be the only group of transposons, which was active in mammals [3]. They occupy a significant part of the genome corresponding to numerous “copy and paste” episodes: over 40% of the human genome is recognized as originated from REs [4,5]. Due to their abundance, REs also inevitably spread in functional genomic regions; for example, they are essential in gene regulation for acting as functional promoters [6] and enhancers [7,8]. As a result, REs could provide new transcription factor binding sites [9,10]. Moreover, they could be involved in chromatin reshaping by making the chromatin transcriptionally active (euchromatin) [11] or inactive (heterochromatin) [12]. Being actively involved in the genome homeostasis, REs evolve under the natural selection as parts of gene regulatory modules. However, given that every de novo RE insertion is not orthologous to the ancestral state, REs can be utilized to estimate relative rates of gene regulatory evolution. Recently we have published a concept that the proportion of functional features held by transposons positively correlates with the rate of regulatory evolution of the respective genes [13]. Furthermore, this information can be aggregated to characterize evolutionary rates of molecular pathways and gene ontology terms with a set of specific quantitative scores [13]. To calculate this score, only exact positions of REs and knowledge of functional features in the region of interest are required. 

In addition to participation in molecular pathways [14,15], individual genes could be aggregated according to apparent common co-expression networks [16] or involvement in complex gene signatures [17]. To study both structural and functional features of gene evolution, we recently proposed analytic pipeline termed RetroSpect. In RetroSpect, the approaches like pathway activation level calculation [18,19] or quantization of gene ontology (GO) clustering features [20,21] are used to characterize high-throughput regulatory impacts of the REs [22,23]. RetroSpect was applied to analyze the impact of RE-linked regulation for all human genes at the level of transcription factor binding sites (TFBS) [13]. REs are known to be important source of TFBS and cryptic promoters for human cells [24]. 

RetroSpect is based on a rationale of measuring the proportion of gene-proximate functional features (e.g., TFBS) hosted by the REs. This proportion is thought to reflect rates of gene regulatory evolution, thus enabling to identify genes with quickly and slowly evolving regulatory modules [13]. For each gene, 10-kb neighborhood around its transcriptional start site (TSS) was analyzed, as TSS-proximal regions are thought to be enriched in key regulatory modules such as promoters and enhancers [25]. Proportions of RE-linked TFBS were calculated based on the published experimental chromatin immunoprecipitation sequencing (ChIP-Seq) data for different human cell lines. Higher number of sequencing reads (=hits) here means stronger binding with transcriptional factors for the same locus, and vice versa [26]. This approach was found effective for data analysis at the levels of both individual genes and molecular pathways [13]. 

The major processes enriched by RE-linked regulation are associated with olfaction, color vision, spermatogenesis and fertilization, some aspects of immune and hormonal responses, intracellular molecular trafficking, metabolism of amino acids, vitamins and fatty acids metabolism, xenobiotic metabolism and detoxication. Quite to the contrary, the deficient pathways were involved in protein synthesis and ribosome biogenesis, RNA transcription and processing, nuclear chromatin organization, cell cycle, apoptosis, cell contacts, embryo development, most of the signaling pathways, cellular stress response, oxidative phosphorylation in mitochondria and some other aspects of immunity. Remarkably, within the top enriched cohort, we found approximately three times higher number of genes known for noncoding RNAs than in the bottom cohort of the same size.

However, functional features other than human TFBS have not been examined in the same manner. In this study, we used the RetroSpect pipeline to study other essential regulatory elements—histone epigenetic marks. We considered five major epigenetic histone modifications (methylation or acetylation) that are involved in gene regulation—H3K4me3, H3K9ac, H3K27ac, H3K27me3 and H3K9me3. In particular, the H3K4me3 modification is highly enriched at active promoters near transcription start sites and considered as transcription activation epigenetic biomarker [27]. Similarly, H3K9ac and H3K27ac modifications denote active gene transcription [28]. On the contrary, H3K27me3 and H3K9me3 are heterochromatin-associated histone marks specific for constitutive and facultative heterochromatin, respectively [29]. The presence of H3K27me3 and H3K9me3, therefore, indicates repressed transcriptional activity in neighboring genome regions. Thus, the abovementioned five epigenetic marks form two functional groups: one associated with transcriptional activation (H3K4me3, H3K9ac and H3K27ac) and one associated with transcriptional suppression (H3K27me3 and H3K9me3).

We ranked regulatory evolution rates of human genes and molecular pathways by using high throughput data on five histone marks from the ENCODE project [30] for five human cell lines. Based on H3K4me3, H3K9ac, H3K27ac, H3K27me3 and H3K9me3 histone tags, respectively, we investigated 25,075 human genes and 3121 molecular pathways. As for the previous TFBS analysis, the processes of “amino acids and polyamines metabolism”, “lipid metabolism”, “detoxication and catabolism of xenobiotics”, “sensory perception and neurotransmission” and “immunity linked pathways” showed signs of the fastest regulatory evolution, while the processes “DNA metabolism and chromatin structure linked pathways”, “nucleic base metabolism” and “translation and protein maturation” demonstrated the slowest regulatory evolution. However, histone tags showed different regulatory patterns and formed two distinct clusters: promoters/active chromatin tags (H3K4me3, H3K9ac, H3K27ac) vs. heterochromatin marks (H3K27me3, H3K9me3). Our findings suggest that in humans, distributions of transposon-linked histone tags evolved in a coordinated manner according to their functional roles.

## 2. Materials and Methods

### 2.1. Quantitative Scores of Genes and Pathways Regulatory Evolution

To evaluate the RE-associated regulatory impact of individual genes, we introduced a quantitative score (Equation (1))—gene RE-linked enrichment score (GRE score) of an individual gene is the sum of RE-specific hits (histone modification marks) found in a 10kb-neighborhood of its TSS, that is normalized by the average content of RE-specific hits for all examined genes. For every gene, GRE enables us to measure its regulation by RE-linked hits. However, this score is limited in a way that different genes with the same GRE value could have a substantially different number of total hits (both RE-linked and not) in their TSS neighborhood, so the GRE score cannot be directly used to compare genes. Therefore, it is required to have another normalized score that shows how the regulatory region of a gene is enriched by RE-linked hits with respect to total hits. To this end, the normalized gene RE-linked enrichment score (NGRE) was proposed. For an individual gene, the NGRE is equal to the ratio of GRE score to balanced total number of hits (not only RE-linked) for the gene [13].

Similar scores were proposed to evaluate the impact of hits at the level of molecular pathways [13] termed pathway involvement index (PII) and normalized pathway involvement index (NPII). PII reflects the total impact of RE-linked hits on the regulation of an individual molecular pathway. The bigger PII is, the stronger impact of RE-linked hits on the overall regulation of a molecular pathway should be, and vice versa. However, PII is not informative enough to estimate RE-linked regulation of a pathway in the context of its total regulation. We, therefore, introduced another score termed NPII (normalized PII) which is needed to estimate the relative RE-linked impact on the regulation of a whole molecular pathway. Higher NPII indicates stronger relative impact of RE-linked TFBS on the general regulation of a molecular pathway, and vice versa [13].

#### 2.1.1. GRE and NGRE Scores

Let us assume that the positions of both RE segments and target histone marks within the chromosomal region associated with a gene *g* are known. Then, for any particular gene *g*, *GRE* score is calculated according to the formula (Equation (1)):(1)$$GRE_{g}=\frac{HES_{g}}{\frac{1}{n}{\sum^{\;}}_{i=1}^{n}HES_{i}},$$
where *GRE_g_* is *GRE* score for a gene *g*; *HES_g_* is the number of RE-linked hits for a gene *g*; *i* is gene index, and *HES_i_* is the number of RE-linked hits for gene *i*; *n* is the total number of genes under investigation. Normalized gene RE-linked enrichment score (*NGRE* score) for a gene *g* is calculated as follows (Equation (2)):(2)$$NGRE_{g}\;=\frac{GRE_{g}}{GHE_{g}}.$$

Here the *GHE* (gene hits enrichment) score characterizes gene-specific hits distribution trends, expressed by the formula (Equation (3)):(3)$$GHE_{g}=\frac{THS_{g}}{\frac{1}{n}{\sum^{\;}}_{i=1}^{n}THS_{i}},$$
where *THS_g_* is the total number of hits mapped in the 10kbp neighborhood of gene *g*, *i* is gene index, *n* is the total number of genes.

#### 2.1.2. PII and NPII Scores

Pathway involvement index (PII) is expressed by the formula (Equation (4)):(4)$$PII_{p}=\frac{\sum_{i=1}^{n}GRE_{i}}{n},$$
where ${\mathrm{PII}}_{p}$ is PII score for a pathway *p*; *GRE_i_* is the GRE score for gene *i*; *n* is the total number of genes in pathway *p*. The normalized pathway involvement index (NPII) is calculated as follows (Equation (5)):(5)$$NPII_{p}\;=\frac{PII_{p}}{PGI_{p}},$$
where *PII_p_* is PII for pathway p; *PGI_p_* is pathway gene-based index for a pathway *p* introduced to assess the impact of total hits (not only RE-linked) on the regulation of molecular pathways. *PGI* for pathway *p* is expressed by the formula (Equation (6)):(6)$$PGI_{p}=\frac{\sum_{i=1}^{n}GHE_{i}}{n},$$
where *GHE_i_* is *GHE* score for gene *i*; *n* is the number of genes in pathway *p*.

### 2.2. Primary Data

From the UCSC Human Genome Browser [31], we downloaded coordinates of human protein-coding genes (RefGenes table, genome assembly hg19). For each gene and each of five human cell lines investigated here (K562, HepG2, GM12878, MCF-7, HeLa-s3), we took all REs overlapping 10kbp region around transcription start site (5kbp upstream and 5kbp downstream). Coordinates of histone modifications—H3K4me3, H3K9ac, H3K27ac, H3K27me3 and H3K9me3—were extracted from the ENCODE database [32].

The reference human genome assembly 2009 (hg19) was indexed by the Burrows–Wheeler algorithm using BWA software, version 0.7.10 [33]. Concatenation of fastq files with single-end or pairwise reads, alignment to the reference genome and filtering were performed by BWA, Samtools (version 1.0), Picard (version 1.92), Bedtools (version 2.17.0) and Phantompeakqualtools (version 1.1) software packages. Aligned reads for histone modification marks for every cell line were mapped on the RE sequences annotated by RepeatMasker (version 3.2.7) and downloaded from the UCSC Browser (RepeatMasker table) [31].

After the calculation of GRE, NGRE, PII and NPII statistics for each gene/pathway and cell line, we calculated the average value of each statistic for genes and pathways across all cell lines under analysis.

### 2.3. Gene Expression Data

From the ENCODE database [34] we obtained RNA sequencing gene expression profiles for human cell lines using the following set of filters: “Transcription”, “total RNA-Seq”, “gene quantifications”. For three out of five cell lines of interest, we found 19 experiments containing gene expression data in two technical replicates: 11 experiments for K562 cell line, five for HepG2 and three for GM12878. Accession numbers are shown in Appendix A.

### 2.4. Enrichment Analysis for Groups of Differential Genes

We performed gene ontology analysis of genes that were either enriched or deficient in epigenetic regulatory marks linked with REs (RRE-enriched and RRE-deficient genes, respectively) by applying DAVID software (version 6.8) [35] using human gene IDs extracted from USCS Genome Browser [31]; RRE stands for “RE-linked regulation”. As the background parameter for the annotation, we took the entire list of genes in the analysis. Our target functional categories were directed GO terms of biological functions, molecular processes and cellular components derived from “Functional Annotation Chart” in DAVID. All the directed terms were extracted with the corresponding enrichment score values. To merge DAVID annotation terms into more general categories, we developed a semi-automatic supervised method. The list of sixteen categories was defined in [13] and contained example terms from each category. To attribute a new term to one of the categories, we found a number of shared genes between the term and each of the example terms. The largest intersection size allowed us to determine the preliminary category of the term. When all terms were split into the categories, we manually checked and fixed misclassification (less than 5% of all cases). As a result, each GO term was assigned to a single category.

To quantify the enrichment of a category for REs-linked regulation, we considered all GO terms in the category and calculated the aggregated *p*-value by Fisher’s method. Obtained *p*-values reflect the enrichment level; however, the use of any significance threshold is not correct in this case. Our approach, like other types of gene set enrichment analysis, has a limitation that the categories are not entirely independent: some genes could correspond to several GO terms. Therefore, *p*-values cannot be adequately corrected for multiple testing. Considering this fact, we analyzed *p*-values guided by the empirical rule: the less *p*-value is, the more enriched the category is.

### 2.5. Measuring Pathway Enrichment by RE-Linked Hits

Gene architecture data of the molecular pathways were extracted from the following databases: BioCarta [36] (downloaded on March 2015), KEGG [37] (downloaded on June 2015), NCI (https://cactus.nci.nih.gov/ncicadd/about.htm) (downloaded on March 2015), Reactome [38] (downloaded on March 2015) and Pathway Central (Pathway Central 2019) (downloaded on March 2015 from http://www.sabiosciences.com/pathwaycentral.php). Data on molecular pathways structure were downloaded in .xml and .biopax formats from these databases and used in our computational algorithm [19]. We calculated PII and NPII scores for 3121 pathways for every cell line. To attribute pathways to sixteen functional categories predefined by [13], we applied the same algorithm as for the classification of GO terms. After the primary classification, we manually checked and corrected the categories; each pathway was assigned to a single category.

Enrichment of the categories after the pathway analysis was calculated in the following way. We calculated the EASE score, which is a modification of Fisher’s exact test [35]. In our case, the EASE score was calculated according to the following contingency table (Table 1). The EASE method provides the *p*-value for each category; however, we considered these values in a ranking-like way, because categories were unlikely to be completely independent.

### 2.6. Combination of Gene- and Pathway-Based Enrichment Scores

To combine the results obtained in two previous pipelines (according to pp. 2.3. and 2.4.), we applied Fisher’s exact test to aggregate the obtained *p*-values. We analyzed the results in a qualitative manner instead of quantitative as it was performed in previous works [9,13] because of the inevitable drawbacks of enrichment analyses.

## 3. Results

### 3.1. Primary Data Analysis

From the ENCODE project database, for five human cell lines, we extracted 200889, 593717, 373112, 302267 and 86197 tags of H3K4me3, H3K9ac, H3K27ac, H3K27me3 and H3K9me3 histone modifications, respectively, that were mapped 5 kbp upstream and 5 kbp downstream of gene transcription start sites. Among them, 28,0%, 20,0%, 22,4%, 20,3% and 29,0% respectively, overlapped with unambiguously mapped human RE sequences. The following five cell lines were examined: K562, GM12878, HepG2, MCF-7, HeLa-s3 (malignant blood—K562 and GM12878, liver, mammary gland and cervix respectively). Then, we employed scores (GRE, NGRE, PII and NPII) that were proposed by us previously to evaluate RE-linked regulation by using transcription factor binding site (TFBS) data [13]. We did not change the analytical formulas of scores, however, we changed their concepts to deal with epigenetic marks instead of TFBS.

For every gene, its GRE score could be used as a measure of the enrichment level by the RE-linked hits. In turn, NGRE score is the normalized GRE on the proportion of total hits (not only RE-linked) overlapping with the gene neighborhood. High NGRE value means stronger impact of RE-specific regulation on overall regulation of a particular gene, and vice versa. The GRE and NGRE scores of 25,075 human genes were calculated in this study.

Similarly, PII score reflects enrichment of the molecular pathways by the RE-linked hits, whereas NPII score was designed to estimate *normalized* impacts of REs in the regulation of molecular pathways. Higher NPII suggests stronger RE-linked regulatory impact for an individual molecular pathway and, consequently, faster evolution of the corresponding pathway regulatory network [13]. The PII and NPII scores of 3121 molecular pathways were calculated.

After computing GRE, NGRE, PII, NPII statistics of each gene/pathway of every cell line, we averaged the values of each statistic for genes and pathways across all cell lines in the study to operate the scores that represent several human tissues simultaneously. All cell line-specific and averaged calculated GRE, NGRE, PII, NPII scores are in Appendix A.

The REs-liked histone marks of interest occupied minor fractions of all chromatin marks. Depending on the cell line, these fractions were 21–31%, 13–29%, 14–33%, 15–26% and 22–35% for *H3K4me3, H3K9ac*, *H3K27ac*, *H3K27me3* and *H3K9me3* tags, respectively (Appendix A). These data are not in line with the previous findings where RE-linked TFBS formed the majority of all TFBS hits [9]. The highest proportions of RE-linked histone marks were made by the representatives of SINE group of REs: 17–25%, 10–24%, 11–28%, 8–14% and 7–14% for the modifications H3K4me3, H3K9ac, H3K27ac, H3K27me3 and H3K9me3, respectively, in different cell lines. For the LINE group of REs they were 9–14%, 5–13%, 6–16%, 6–13% and 11–19%, respectively. Finally, for the LTR retrotransposons and endogenous retroviruses (LR/ERVs) the proportions were 3–4%, 2–4%, 2–6%, 3–6% and 7–13%, respectively. However, it should be noted that the number of histone modification tags mapped on all retroelement classes is not equal to sum of tag numbers separately mapped on LINEs SINEs and LR/ERVs because one tag could overlap with several individual REs [9]. Previously, Estecio et al. and coauthors found that increased content of B1 SINE elements in rodent gene promoters was connected with the repressing histone marks [39]. In this study we found no linkage between human gene-proximate SINE contents and H3K27me3 or H3K9me3 histone modifications.

### 3.2. Correlation between Histone Tags Based on GRE/NGRE and PII/NPII Scores

First, we analyzed how profiles of histone tags vary within cell lines. For this purpose, we obtained 25 profiles from the ENCODE database (5 tags for 5 cell lines), calculated correlations for each pair of profiles and applied the biclustering on the square symmetric matrix of correlations (Figure 1). For each histone tag, we found that obtained profiles were highly congruent across cell lines, and the tissue-specific component had only a minor impact on the profiles. Therefore, in further analysis, we averaged values of GRE, NGRE, PII, and NPII scores across five cell lines of interest.

In addition, we observed that the histone marks formed two clear-cut groups with strongly correlated gene distribution profiles for their members (Figure 1). One group contained promoter and active/open chromatin marks (H3K4me3, H3K9ac, H3K27ac), whereas another group contained inactive (constitutive and facultative heterochromatin, respectively) marks H3K9me3 and H3K27me3.

The different functionalities of those two groups were clearly illustrated by correlations of their histone profiles with the gene expression data obtained for the same cell lines (Figure 2). To this end, we extracted available RNA sequencing profiles from the ENCODE project repository. Totally, six profiles were available for cell line GM12878, ten for cell line HepG2 and twenty-two for cell line K562. For the MCF-7 and HeLa-s3 cell lines, the were no RNA sequencing data available. In this study, we did not consider microarray hybridization data because RNA sequencing is thought to provide more accurate data being currently the gold standard approach in high throughput transcriptomic research [40,41]. We observed a clear trend that the profiles for active/open chromatin marks positively correlated with the gene expression. In contrast, the inactive (constitutive/facultative) heterochromatin marks showed negative correlations with the expression profiles (Figure 2). This confirmed that the different histone modification tags presented in the ENCODE project database were related to their expected molecular functions.

We then analyzed GRE, NGRE, PII and NPII metrics for the different histone modifications. We visualized the scatterplots for each pair of histone marks and rediscovered two clear-cut groups that display strongly correlated scores (Figure 3). As previously, histone modifications became divided into two groups: the active/open chromatin marks (*H3K4me3*, *H3K9ac*, *H3K27ac*) and the constitutive/facultative heterochromatin marks (*H3K9me3*, *H3K27me3*).

### 3.3. Genes and Molecular Pathways Enriched or Deficient in RE-Linked Regulation

Then, the dependencies in pairs GRE/NGRE and PII/NPII were analyzed to identify genes and pathways, respectively, that have their regulation enriched or deficient by RE-linked histone tags (RRE-enriched and RRE-deficient genes/pathways). For that purpose, 25075 human genes and 3121 pathways in analysis were examined respectively on scatter plots with X axis showing GRE score for genes or *PII* for pathways and Y axis showing NGRE score for genes and NPII for pathways. These scatterplot allow us to identify genes/pathways that have either high or low RRE impact (Figure 4). Within each scatterplot we fitted a one-parameter linear regression (y = ax) and took the outlier points: 5% of points above the trend line and 5% of points below the trend line (5% of total number of points) by using the Euclidean distance to the regression line. Bottom and top outliers we denoted as RRE-enriched or RRE-deficient, respectively (Figure 4).

### 3.4. Gene Ontology (GO) Annotation of Top RRE-Enriched and Deficient Genes

We performed Gene Ontology (GO) annotation using DAVID software for ten groups of genes (top and bottom genes for five human histone tags) and obtained *p*-values of EASE enrichment score for each of 826 GO direct terms. Then, we aggregated GO terms into sixteen broader functional categories formulated according to our previous article [13]. For this purpose, we combined *p*-values for terms corresponding a particular category by Fisher’s method. The heatmap representation of the obtained aggregated *p*-values for all functional categories is shown in Figure 5.

The two previously identified groups of histone marks corresponding to active or inactive chromatin were notably distinguishable among the data for functional categories. Analysis of *p*-values revealed the common enriched categories within each group. For example, for the group of “active chromatin” histone modifications, the categories “lipid metabolism”, “electron transfer chain reactions”, “catabolism of xenobiotics” show RRE-upregulation (high number of RRE-enriched genes). In the same group, categories “cytoskeleton organization and cell adhesion linked pathways”, “DNA metabolism and chromatin structure linked pathways”, “nucleic base metabolism”, “translation and protein maturation” and “RNA synthesis” show RRE-downregulation (high number of RRE-deficient genes). The categories “mitochondria linked pathways” and “infection” show ambiguous trends and no significant up- or downregulation could be detected for six remaining categories.

In the “inactive chromatin” group, categories “RNA synthesis”, “translation and protein maturation”, “nucleic base metabolism”, “intracellular signaling”, “mitochondria linked pathways”, “electron transfer Chain Reactions”, “DNA metabolism and chromatin structure linked pathways”, “cell cycle regulation and apoptosis” and “amino acids and polyamines metabolism” were enriched for the RRE-enriched genes. Other categories showed either ambiguous trends or no significant up- or downregulation Figure 5).

Remarkably, we also found a sort of anti-phase manner in enrichment between the two groups of histone tags (Figure 5). For example, categories “infection”, “DNA metabolism and chromatin structure linked pathways”, “translation and protein maturation”, “nucleic base metabolism”, “mitochondria linked pathways”, and “RNA synthesis” were enriched among the RRE-enriched genes in group 2 but at the same time among the RRE-deficient genes of group 1. In contrast, the categories “electron transfer chain reactions”, “infection”, “catabolism of xenobiotics” and “cytoskeleton organization and cell adhesion linked pathways” were enriched among RRE-enriched genes of both functional groups of histone modifications.

### 3.5. Top RRE-Enriched and Deficient Molecular Pathways

RRE-enriched and -deficient molecular pathways were aggregated into the same sixteen categories for all types of histone modifications investigated. As before, the two functional groups of modifications showed specific and clearly distinct enrichment trends (Figure 6).

However, five categories showed similar enrichment trends in both groups: “carbohydrates metabolism”, “electron transfer chain reactions”, “catabolism of xenobiotics”, “cell cycle regulation and apoptosis” and “cytoskeleton organization and cell adhesion”. In contrast, there were six oppositely regulated categories: “perception and neurotransmission”, “RNA synthesis”, “translation and protein maturation”, “lipid metabolism”, “immunity linked pathways”, “DNA metabolism and chromatin structure linked pathways”. For the remaining five categories, the trends were ambiguous or vague for the two compared groups (Figure 6).

### 3.6. Combined Analysis of Gene and Pathway Level Trends

For the majority of the functional categories investigated the gene- and pathway-based analytic pipelines gave yielded results. To combine both types of analyses (gene GRE/NGRE statistics and pathway PII/NPII statistics), we applied Fisher’s method to the obtained *p*-values of gene/pathway enrichment in the categories (Figure 5 and Figure 6) and visualized the resulting *p*-values in the same manner (Figure 7). The significance threshold was not applicable for this comparison due to the high complexity of the multi-step statistical analysis used here. We focused on the trends that could be observed for the different histone groups (Figure 5). The two previous groups of histone tags consistently showed largely different coordinated RRE profiles.

We found two out of 16 molecular categories enriched by RE regulation in the first group associated with the active/open chromatin while not affected in the second group linked with condensed/inactive chromatin. Specifically, these were “lipid metabolism” and “carbohydrates metabolism” categories. The category “cytoskeleton organization and cell adhesion” was enriched in the first group but showed a contradictory trend in the second group. Three categories were enriched in both groups of histone tags: “electron transfer chain reactions”, “catabolism of xenobiotics”, “aminoacids and polyamines metabolism”. Two categories were enriched in the first group but deficient in the second: “perception and neurotransmission” and “immunity linked pathways”. Four categories were deficient in the first group but enriched in the second group: “RNA synthesis”, “translation and protein maturation”, “DNA metabolism and chromatin structure linked pathways” and “cell cycle regulation and apoptosis”. Two categories showed baseline RRE in the first group but were enriched in the second group: “Infection” and “intracellular signaling”. One category (mitochondria linked pathways) was oppositely regulated in the first group and upregulated in the second group. Finally, one category showed unclear trends in both groups: “nucleic base metabolism”.

As a result, our data showed three similarly and six oppositely regulated categories between the two groups, thus suggesting significant differences between RRE of active and inactive chromatin marks. Notably, we found three categories that were RRE enriched in both groups, but not a single category that would be deficient in both groups. Overall, these results are consistent with our previous findings made on TFBS data that the above sixteen functional categories are strongly regulated by REs [9].

## 4. Discussion

In this study, we examined high throughput RE-linked features of gene regulation by histone modifications. One of the major functions of epigenetic regulation of gene expression is thought to be the control of transposable elements, most frequently their repression [42]. Here we performed a systematic analysis of the reciprocal influence of transposable elements on function and evolution of human epigenetic mechanisms. In many previous reports, influence of transposable elements on the development of epigenetic regulatory networks has been documented. For example, as learned from the analysis of evolution of duplicated genes in human DNA, the key factor influencing the regulatory epigenetic landscape is the presence of transposable elements [43].

In this study, we analyzed quantitative characteristics of gene regulation associated with RE-linked histone marks. Another class of transposable elements, DNA transposons, constitute significantly smaller fraction than REs of only up to 3% of the human genome and were most likely not active after mammalian radiation [4]. However, they also impact regulation of gene expression. For example, the human genome contains several thousand copies of the HsMar1 element TIRs that influence the expression of neighboring genes through epigenetic regulation [44,45,46]. Moreover, these elements also contain a functional silencer influencing expression of human gene located nearby [47]. However, the analytic value of DNA transposons in RetroSpect pipeline for human genome is limited as they are not numerous and most of known human genes lack their inserts near TSS. This makes relative measures of regulatory enrichment problematic for both individual genes and molecular pathways. We, therefore, focused on the RE-linked regulation of gene expression, as REs are more abundant and mostly represent more recent inserts than DNA transposons. However, in applications of RetroSpect pipeline to non-mammalian species, DNA transposons may become useful tags in case of their high content and recent insertional activities in the genomes of interest.

We used molecular data obtained from ENCODE project for five human cell lines; 1,556,182 histone tags of all types were investigated in total. Five types of histone marks of open/active or condensed/inactive chromatin were studied: H3K4me3, H3K9ac, H3K27ac and H3K27me3, H3K9me3, respectively. The gene-based characteristics were further aggregated into quantitative scores for the molecular pathways. This allowed us to identify top differential genes and molecular pathways enriched of deficient in regulation by RE-associated histone tags.

We worked with the human cell lines instead of normal tissues due to public availability of high-throughput profiles for target histone marks and RNA sequencing data for the former. The five cell lines selected for our analysis represented different tissues of human body. However, we found that the histone tags highly correlated between the different cell lines, thus suggesting only minor impact of a tissue-specific component on the data. However, availability of novel epigenetic datasets corresponding to normal human tissues would be extremely desirable for further re-analysis of data with RetroSpect pipeline.

For each histone mark at the gene-based way, the analyses yielded two sets of genes (RRE-enriched and deficient) which were functionally annotated with GO direct terms. We developed the semi-automatic annotation algorithm and obtained the list of 826 GO terms attributed further to sixteen functional categories. The same procedure was applied to molecular pathway analysis, and 3121 pathways were attributed to the same sixteen categories.

These data were used to interrogate RRE enrichment or deficiency in the sixteen functional categories previously identified as strongly differential in RE regulation according to transcription factor binding sites data [13]. Interestingly, two functionally different groups of histone marks showed markedly different RRE patterns of the above sixteen functional categories. This result confirms the coordinated behavior of histone modifications associated with gene expression variation [48] and specifies this state that RE-linked histone modifications are correlated within “active” and “repressive” groups of marks.

The first group representatives (H3K4me3, H3K9ac, H3K27ac) showed RRE patterns highly congruent with previous observations of TFBS data, e.g., they were enriched in categories of “amino acids and polyamines metabolism”, “lipid metabolism”, “detoxication and catabolism of xenobiotics”, “sensory perception and neurotransmission” and “immunity linked pathways”; at the same time, they were deficient in “DNA metabolism and chromatin structure linked pathways”, “DNA metabolism and chromatin structure linked pathways”, “nucleic base metabolism” and “translation and protein maturation”.

The second group of histone marks showed clearly more distinct RRE trends with the previous TFBS data. Therefore, we hypothesize that the RE-linked histone marks with the same meaning (activation or inactivation of chromatin) possibly have their evolution coordinated, although the marks of different functionality probably display the opposite evolutionary trends in many functional categories. However, two functional categories that were enriched in both groups of histones here were also RRE-enriched according to previous TFBS data investigations [9,13]: “electron transfer chain reactions”, “catabolism of xenobiotics” and “aminoacids and polyamines metabolism”.

Our data suggest that histone modifications demonstrate more complex trends of regulation by REs than TFBS. Therefore, further investigation of functional genomic marks and their direct comparisons are needed to uncover the high-throughput impact of REs on human gene regulation.

## Figures and Tables

**Figure 1 cells-08-01034-f001:**
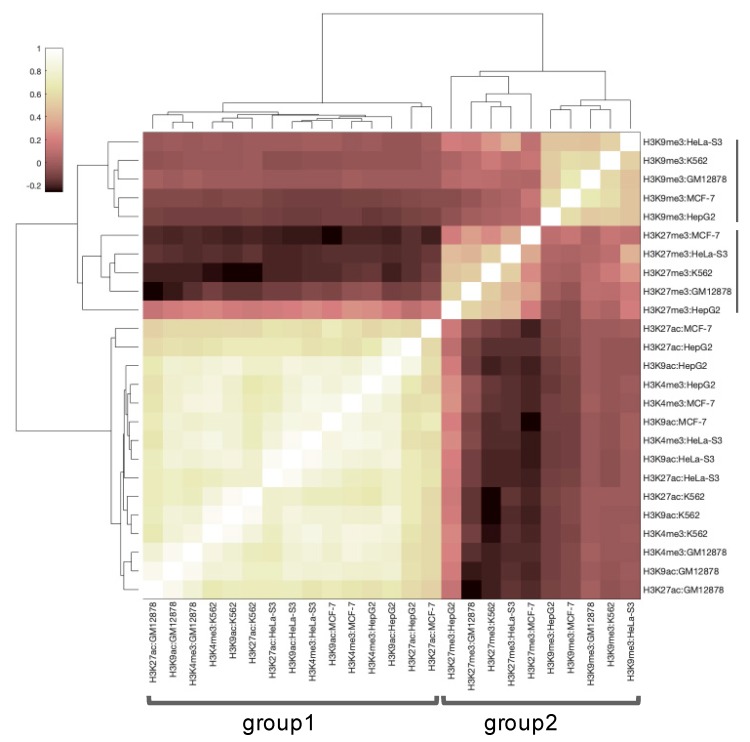
Correlation between profiles of five histone tags (*H3K4me3, H3K9ac*, *H3K27ac*, *H3K27me3* and *H3K9me3*) in five human cell lines (K562, GM12878, HepG2, MCF-7, HeLa-s3).

**Figure 2 cells-08-01034-f002:**
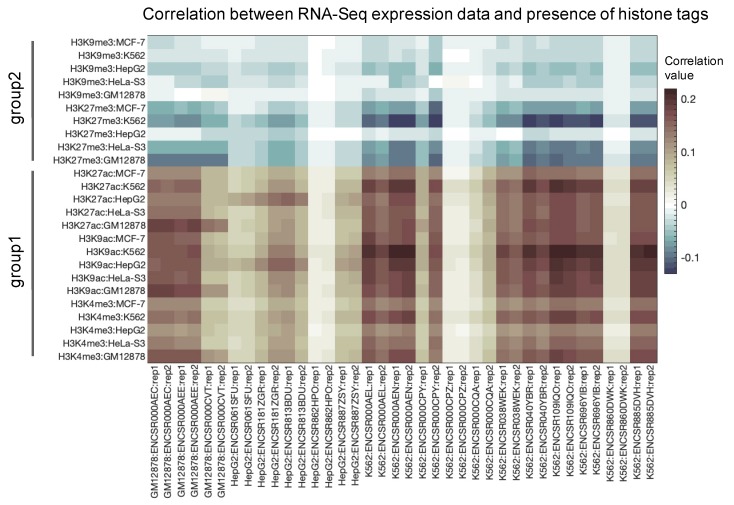
Correlations of histone profiles with the RNA sequencing gene expression profiles available in ENCODE database.

**Figure 3 cells-08-01034-f003:**
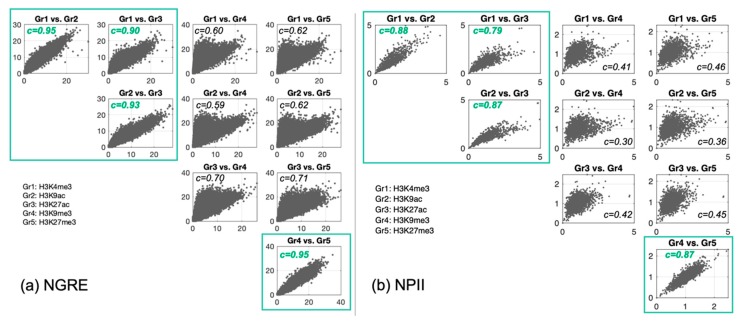
Scatterplots of (**a**) (normalized gene RE-linked enrichment score (NGRE) against NGRE) and (**b**) (normalized pathway involvement index (NPII) against NPII) for five histone tags in *Homo sapiens*. Each dot represents a single gene. Correlation values are denoted with *c* and showed within each subplot; all values were statistically significant (*p*-value < 0.001). Green squares highlight highly correlated histone tags (*c* > 0.75).

**Figure 4 cells-08-01034-f004:**
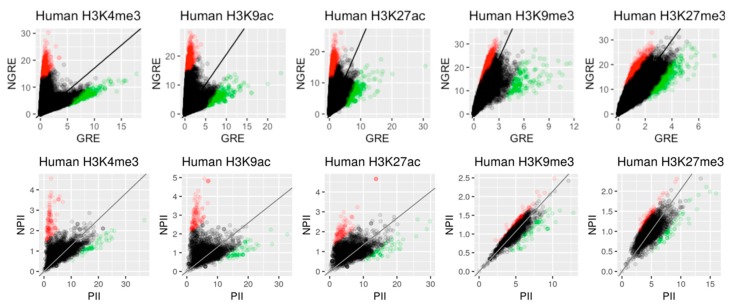
Scatterplots of (NGRE against GRE) and [NPII against PII] for five histone tags in *H. sapiens*. Each dot represents a single gene. Red and green colours represent genes/pathways enriched and deficient (respectively) for retroelements (RE)-linked epigenetic regulation.

**Figure 5 cells-08-01034-f005:**
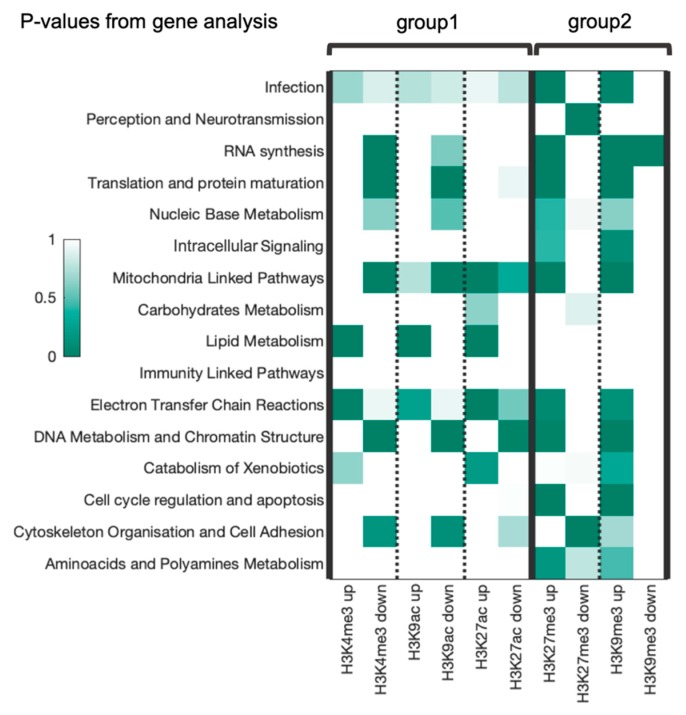
The *p*-values heatmap for association with regulatory marks linked with REs (RRE) up- or downregulation for each category for each histone tag. Up/down means enrichment/deficiency, respectively.

**Figure 6 cells-08-01034-f006:**
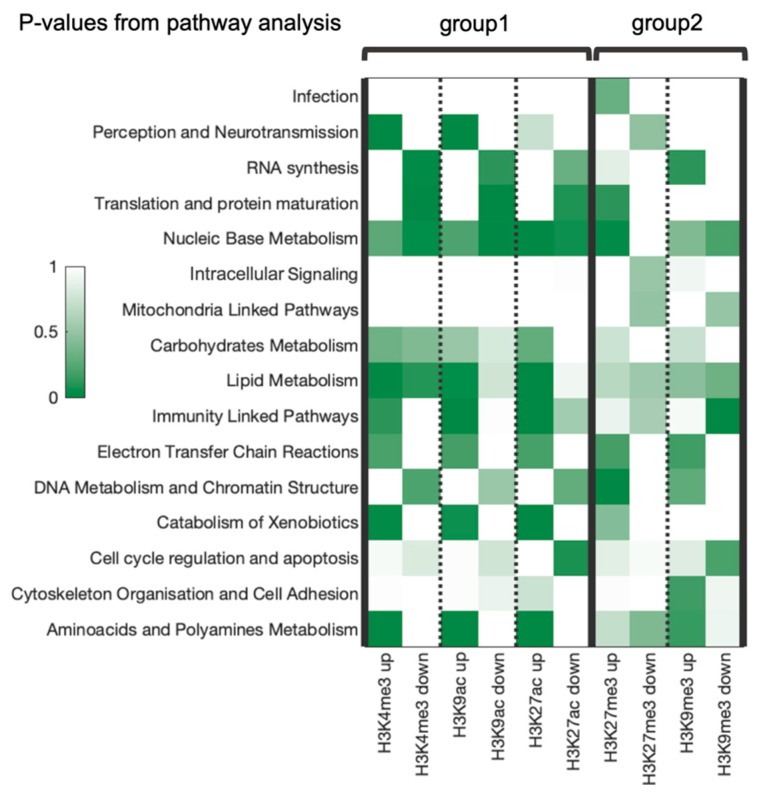
EASE Scores (modified Fisher exact *p*-values) heatmap for association with RRE up- or down regulation for each functional category for five histone tags under study. Up/down means enrichment/deficiency, respectively.

**Figure 7 cells-08-01034-f007:**
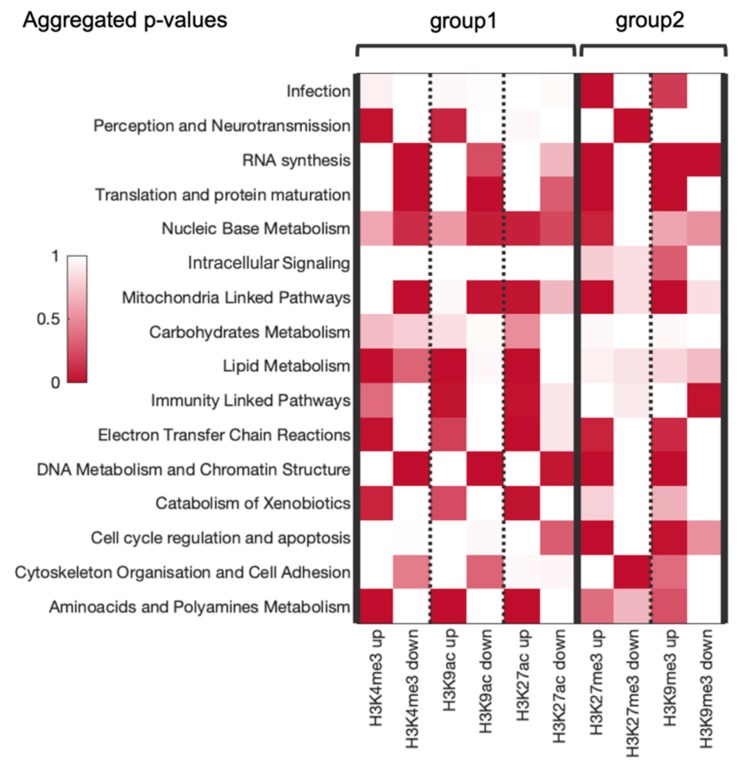
Heatmap for the combined analysis of RRE enrichment or deficiency for two groups of histone tags. Up/Down means enrichment/deficiency, respectively. Each cell represents a single *p*-value.

**Table 1 cells-08-01034-t001:** Contingency table for the EASE score. X is the number of enriched/deficient pathways in the category, Y is the number of all enriched/deficient pathways, Z is the number of all pathways in the category, and K is the number of all pathways analyzed.

Differential Regulation	Number of Pathways in Category	Total Number of Pathways
**Number of Enriched/Deficient Pathways**	max (0, X-1)	Y
**Number of not Enriched/Deficient Pathways**	Z-X	K-Y

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
