# Peer review of "H3K4me3, H3K9ac, H3K27ac, H3K27me3 and H3K9me3 Histone Tags Suggest Distinct Regulatory Evolution of Open and Condensed Chromatin Landmarks"

_cells, 2019, doi:10.3390/cells8091034_

Round 1

Reviewer 1 Report

The manuscript by Igolkina and collaborators describes the contribution of TEs in shaping the human gene regulatory pattern.

Although the methods are solid and the results well-presented the main limitation of this study is that only RTE insertions have been considered in the 10Kbp DNA window around the TSS. In the human genome DNA transposons constitute up to 3% of the genome (Lander et al. 2001, doi:10.1038/35057062). although their frequency is far less than retrotransposons, and the number of active families is very small, their contribution to the regulation of endogenous gene expression is not negligible. As an example, the human genome contains several thousand copies of the HsMar1 element TIRs that could affect the expression of neighboring genes by epigenetic regulation (Renault et al., 2019, doi:10.1186/s12863-019-0719-y). The influence of DNA methylation at the Hsmar1 sites have been previously described by Jursch et al., 2013 (doi: 10.1186/1759-87) and could also be the key to explain some peculiar features of its promoter (Palazzo et al., 2019 doi:10.1186/s13100-019-0155-6). Moreover, it has been shown that Hsmar1 elements contain a silencer, which can also affect gene expression (Bire et al., 2016 doi:10.1371/journal.pgen.1005902). Therefore, the contribution of DNA transposons cannot be disregarded to provide a complete description of the contribution of TEs in the evolution of human regulatory regions.

OTHER MAJOR COMMENTS
The state of the art is poorly described in my opinion. A wider presentation of the epigenetic contribution of TEs to host gene expression should be done based on previously published papers.

Only RTEs are described. Although very few Class II element families are present in the human genome a description should be given in the introduction.

A representation of the occurrence/frequencies of transposable elements found in the analyzed genomic regions is missing and should be provided to give the readers a better idea of their presence in gene control regions.

How the results obtained correlate with the actual expression of the genes analyzed? In other words, are genes enriched in activating markers expressed/strongly expressed? are genes enriched in repressing markers silenced? This could be easily checked using expression databases and could add value to the work.

What could be said if genes enriched in repressive markers are expressed or highly expressed (or vice versa)? It has been recently shown (Marsano et al., 2019 doi:10.1016/j.tig.2019.06.002) that at least in heterochromatin domains the epigenetic status of genes therein not always obviously correlates with their expression level. This is an additional aspect that the Authors could discuss.

The title is a bit misleading as it says "...histone tags indicate regulatory evolution rates of...". Actually, in the main text there is no mention of either evolutionary analysis or evolutionary rate calculation. Please consider changing the title with a most straightforward one.

The Discussion section should be completely re-written. This section appears more like a summary of results rather than a discussion.
The Author could consider discussing their data in comparison with the results obtained in previous works, such as in (not an exhaustive list!):
- Lannes et al., 2019 (doi:10.3390/genes10030249)
- Jiang and Upton 2019 (doi:10.1186/s13100-019-0158-3)
- Estecio et al 2012 (doi:10.1158/1541-7786)

Authors could also consider my comments above to re-built the discussion section-

MINOR CHANGES SUGGESTED

line 98 "biomarker" I would change to "epigenetic marker"

lines 399-400 remove "Both annotation pipelines are available at GitHub repository" Materials and Methods is the best place for this kind of sentences.

Author Response

The authors are thankful to Reviewer 1 for his/her useful suggestions that hopefully allowed us to improve the manuscript

Item-by-item replies to the Referee comments:

(1)

“ Although the methods are solid and the results well-presented the main limitation of this study is that only RTE insertions have been considered in the 10Kbp DNA window around the TSS. In the human genome DNA transposons constitute up to 3% of the genome (Lander et al. 2001, doi:10.1038/35057062). although their frequency is far less than retrotransposons, and the number of active families is very small, their contribution to the regulation of endogenous gene expression is not negligible. As an example, the human genome contains several thousand copies of the HsMar1 element TIRs that could affect the expression of neighboring genes by epigenetic regulation (Renault et al., 2019, doi:10.1186/s12863-019-0719-y). The influence of DNA methylation at the Hsmar1 sites have been previously described by Jursch et al., 2013 (doi: 10.1186/1759-87) and could also be the key to explain some peculiar features of its promoter (Palazzo et al., 2019 doi:10.1186/s13100-019-0155-6). Moreover, it has been shown that Hsmar1 elements contain a silencer, which can also affect gene expression (Bire et al., 2016 doi:10.1371/journal.pgen.1005902). Therefore, the contribution of DNA transposons cannot be disregarded to provide a complete description of the contribution of TEs in the evolution of human regulatory regions”

… “Only RTEs are described. Although very few Class II element families are present in the human genome a description should be given in the introduction”

Answer: We agree. To make this point clear, we significantly expanded Discussion section, where we discussed the papers mentioned by the Referee and other relevant papers on this topic. We also explained that since DNA transposons occupy minor fraction of the genome, we were unable to detect their presence in 10-kb neighborhood of TSS of most human genes. We therefore restricted the analysis to retrotransposon genomic inserts.

(2)

“The state of the art is poorly described in my opinion. A wider presentation of the epigenetic contribution of TEs to host gene expression should be done based on previously published papers”

… “The Discussion section should be completely re-written. This section appears more like a summary of results rather than a discussion. The Author could consider discussing their data in comparison with the results obtained in previous works, such as in (not an exhaustive list!):

- Lannes et al., 2019 (doi:10.3390/genes10030249); Jiang and Upton 2019(doi:10.1186/s13100-019-0158-3); Estecio et al 2012 (doi:10.1158/1541-7786)”.

Answer: As suggested, we re-wrote Discussion section and discussed the findings highlighted by the Referee.

 (3)

“A representation of the occurrence/frequencies of transposable elements found in the analyzed genomic regions is missing and should be provided to give the readers a better idea of their presence in gene control regions”.

Answer: We agree. Representations of the different groups of transposable elements is now given in Supplementary Table 2 and discussed in Results section (subsection “Primary data analysis”).      

(4)

“How the results obtained correlate with the actual expression of the genes analyzed? In other words, are genes enriched in activating markers expressed/strongly expressed? are genes enriched in repressing markers silenced? This could be easily checked using expression databases and could add value to the work.

…What could be said if genes enriched in repressive markers are expressed or highly expressed (or vice versa)? It has been recently shown (Marsano et al., 2019 doi:10.1016/j.tig.2019.06.002) that at least in heterochromatin domains the epigenetic status of genes therein not always obviously correlates with their expression level. This is an additional aspect that the Authors could discuss.”.

Answer: As recommended by the Reviewer, we compared densities of epigenetic marks in 10-kb TSS window with the expression levels of the corresponding genes established by RNA sequencing in ENCODE project for the same cell lines as epigenetic profiling. Text and new figure were added (Fig.2 of the Revised Manuscript) describing the results suggesting an overall positive correlation for the open chromatin marks and negative correlation – for the condensed chromatin marks. Our results, therefore, fall within a major paradigm of a positive/negative effects of open/condensed chromatin on gene expression. These results also confirm adequacy of the primary data extracted from the ENCODE project database.  

(5)

“The title is a bit misleading as it says "...histone tags indicate regulatory evolution rates of...". Actually, in the main text there is no mention of either evolutionary analysis or evolutionary rate calculation. Please consider changing the title with a most straightforward one”.

Answer: We changed the title to the following: “H3K4me3, H3K9ac, H3K27ac, H3K27me3 and H3K9me3 histone tags suggest distinct regulatory evolution of open and condensed chromatin landmarks”.

(6)

“MINOR CHANGES SUGGESTED

line 98 "biomarker" I would change to "epigenetic marker"

lines 399-400 remove "Both annotation pipelines are available at GitHub repository" Materials and Methods is the best place for this kind of sentences.”.

Answer: Done.

Reviewer 2 Report

Authors of the manuscript performed profiling of human genes and molecular pathways affected by activating and inhibitory histone tags linked to transposable elements involved in gene regulation (5kb upstream and 5kb downstream of transcription start site) using the ENCODE project histone modification data for five different human cancer cell lines. The manuscript provides useful guide for functional analysis of RE linked histone marks but the choice of input data from cancer cell lines raises concerns regarding interpretation and applicability of the results. Therefore, although depending on the scope of this edition, in terms of scientific rigour the study could be considered for publication, there are however some minor and major concerns that would need to be addressed to raise the quality of the manuscript sufficiently: Is there particular reason for using only cancer cell line data? What about primary cells/tissues? Aberrant epigenetic modifications are hallmark of carcinogenesis and they continue to change during tumor progression and metastasis. This should be included in discussion. It is also not clear why data from 5 different cancer cell lines were pooled together. As pointed by authors each cell line derives from different tissue of origin, each characterized by specific methylation pattern. Where there significant differences in PII and NPII statistics between cancer cell lines? 

Does RRE stands for RE-linked regulation? It should be clearly stated.

Line 113 of Introduction, "DNA Metabolism and Chromatin Structure Linked Pathways" is duplicated. 

In Fig 5. description there is duplication of "Up/Down means enrichment/deficiency , respectively."

Author Response

We thank the Reviewer 2 for his/her thorough reading of the manuscript, its positive evaluation  and helpful advises.

Item-by-item replies to the Referee comments:

(1)

“ Therefore, although depending on the scope of this edition, in terms of scientific rigour the study could be considered for publication, there are however some minor and major concerns that would need to be addressed to raise the quality of the manuscript sufficiently: Is there particular reason for using only cancer cell line data? What about primary cells/tissues? Aberrant epigenetic modifications are hallmark of carcinogenesis and they continue to change during tumor progression and metastasis. This should be included in discussion. hange during tumor progression and metastasis. This should be included in discussion. It is also not clear why data from 5 different cancer cell lines were pooled together. As pointed by authors each cell line derives from different tissue of origin, each characterized by specific methylation pattern. Where there significant differences in PII and NPII statistics between cancer cell lines?”

Answer: We worked with the human cell lines instead of normal tissues due to public availability of high-throughput profiles for target histone marks and RNA sequencing data for the former. The five cell lines selected for our analysis represented different tissues of human body. However, we found that the histone tags highly correlated between the different cell lines, thus suggesting only minor impact of a tissue-specific component on the data. However, availability of novel epigenetic datasets corresponding to normal human tissues would be extremely desirable for further re-analysis of data with RetroSpect pipeline. The Revised version of manuscript was updated with these statements (Results and Discussion sections) and a new figure (Fig.1 of the Revised Manuscript) illustrating high congruence between the epigenetic profiles of the five cell lines investigated was added. For each histone tag, we found that obtained profiles were highly congruent across cell lines, and the tissue-specific component had only a minor impact on the profiles. Therefore, in further analysis, we averaged values of GRE, NGRE, PII, and NPII scores across five cell lines of interest.

(2)

“ Does RRE stands for RE-linked regulation? It should be clearly stated.

Line 113 of Introduction, "DNA Metabolism and Chromatin Structure Linked Pathways" is duplicated.

In Fig 5. description there is duplication of "Up/Down means enrichment/deficiency , respectively."”

Answer: Corrected as suggested by the reviewer.

Round 2

Reviewer 1 Report

The manuscript has been largerly improved with the addition of the the results from new analyses that have been performed.

I suggest to include a description of the methods applied to obtain such new data.

A list of the accession number of the RNA-seq profile obtained from ENCODE should be also provided.

Other minor issues.

line 58. "... making the chromatin active... " I would change to "making the chromatin trnascriptionally active "

Figure 1 legend. "... in five less lines " is this correct?

Author Response

The authors are thankful to Reviewer 1 for his/her positive evaluation of the revised manuscript and useful comments

Item-by-item replies to the Referee comments:

(1)

“ I suggest to include a description of the methods applied to obtain such new data”

Answer: A new subsection was added to Materials and Methods

“2.3 Gene expression data

From the ENCODE database [34] we obtained RNA sequencing gene expression profiles for human cell lines using the following set of filters: “Transcription”, “total RNA-Seq”, “gene quantifications”. For three out of five cell lines of interest, we found 19 experiments containing gene expression data in two technical replicates: 11 experiments for K562 cell line, 5 - for HepG2 and 3 – for GM12878. Accession numbers are shown in Supplementary Table 1.”

(2)

“A list of the accession number of the RNA-seq profile obtained from ENCODE should be also provided”.

Answer: A new supplementary table listing accession numbers was added to the manuscript (Supplementary table 1 of the revised manuscript; the previous Supplementary tables 1 and 2 are now Supplementary tables 2 and 3, respectively.

 (3)

“line 58. "... making the chromatin active... " I would change to "making the chromatin trnascriptionally active ”.

Answer: Done.      

(4)

“Figure 1 legend. "... in five less lines " is this correct?”

Answer: we meant “human cell lines”. Corrected in the revised version of manuscript.